# Fish Consumption Behaviour and Perception of Food Security of Low-Income Households in Urban Areas of Ghana

**Edward E. Onumah** [1,*] , **Elizabeth A. Quaye** [2], **Anderson K. Ahwireng** [3] **and Benjamin B. Campion** [4]

1   Department of Agricultural Economics and Agribusiness, University of Ghana, Legon, Accra LG68, Ghana
2   Ministry of Fisheries and Aquaculture Directorate, Accra GP630, Ghana; ankpafoley1@yahoo.com
3   Department of Geography, Planning and International Development Studies, University of Amsterdam, 1018 MV Amsterdam, The Netherlands; andersonkwasi@yahoo.com
4   Department of Fisheries and Watershed Management, Kwame Nkrumah University of Science and Technology, Kumasi AK-039-5028, Ghana; bbcampion@gmail.com
*   Correspondence: eeonumah@ug.edu.gh; Tel.: +233-542835782

**Abstract:** This paper assesses fish consumption behaviour and perception of fish food security of low-income households using three season survey data from 839 interviews in Ghana. The study profiles the types of fish consumed and employs a modified Cobb–Douglas function to examine the determinants of household expenditure on fish consumption, whilst adopting a 1–5 Likert scale to analyze the perception of fish food security. The results confirm that poor households prefer cheaper and small pelagic fish. The mean expenditure on fish consumption per week is estimated to be GHS 31.15 (Euro 4.94 ≅ 0.16). Additionally, it is demonstrated that marital status, religion, occupation, proximity to local market, and city of residence have a positive and significant influence, whilst level of income, seasonality of fish, and the interaction of religion and seasonality of fish demonstrate a negative and significant influence on fish expenditure. Finally, the paper reveals that the majority of households have the perception that fish is readily available and can be obtained throughout the year in good quality. However, households have varied opinions on accessibility of fish. The paper recommends that the government should support and enhance the value chains of small pelagic fish species since they are preferred by poor households.

**Keywords:** small pelagics; three season survey data; availability; accessibility; utilization; stability

## 1. Introduction

The fisheries sector is one of the key sectors supporting the socio-economic development of Ghana. Quagrainie and Chu [1] note that the sector generates about US$ 1 billion in revenue each year and supports about 135,000 fishers in the marine sub-sector alone. Taking a broader perspective, Ghana's fisheries contribute about 5% to annual GDP and support the livelihoods of over 2.6 million Ghanaians [2], and thus help in poverty reduction across the country. Furthermore, Ghana has a vibrant fish-eating culture, and fish is estimated to provide around 60% of the animal protein requirements for both the poor and rich. In addition, fish provides micronutrients essential for human life [3]. The annual per capita consumption of fish in Ghana is currently estimated to be 25 kg, which is higher than the estimated averages of 18.9 kg and 10.5 kg for the world and Africa, respectively [2].

Local fisheries, however, are unable to meet domestic demand possibly due to logistical challenges and overexploitation of the marine stock [4]. To maintain the current per capita levels of consumption,

Ghana currently imports about 50% of fish and fish products to supplement its total national catch [5]. The main sources of fish in the country are marine fisheries, which connect to supply chains from the coast to consumers in the hinterland, and inland fisheries which originate in Ghana's rivers and lakes, predominantly the Lake Volta [4]. Fish farming is also progressively gaining grounds in contributing to the domestic market. However, farmed fish is relatively expensive and may be unaffordable to poor consumers [6]. Quagrainie and Chu [1] revealed that the poor are known to depend largely on low-priced small pelagic fish species, such as sardines and mackerel, which are processed in a variety of ways (dried, smoked, fried, etc.), for their protein needs because they are available, affordable, and easily accessible all year round for preparation of meals at home.

Decisions on the type of fish and how much to purchase and consume are believed to be affected by various factors. Fish consumption levels, frequency and food budget allocation could be influenced by socio-economic and geographic characteristics of consumers and by fish attributes [7–9]. A study by the Bank of Ghana [10] reveals that households in Ghana spend about 22.4% of their food budget on fish consumption while poor households allocate about 25.7%. However, knowledge of factors influencing this allocation is still limited.

This paper analyzes the contribution of low-cost fish to the food and nutrition security of low-income consumers in two of Ghana's cities. Specifically, it profiles the types of fish consumed and analyzes the factors influencing income allocation to fish consumption. It also highlights the perception of fish food and nutritional security by poor consumers in terms of availability, accessibility, utilization, and stability [11]. The paper is based on a three-season survey carried out in low-income neighbourhoods of a coastal city (Accra) and an inland city (Tamale), thereby offering opportunities for comparing the flow and consumption patterns of fish in disparate geographical settings, whilst taking into consideration the influence of seasonality on fish prices. Thus, the study fulfils an identified need to analyze the degree of influence of different factors on income allocation to fish consumption by low-income households. Further, it adopts an unconventional approach to examining food and nutrition security from the perspective of consumers with a focus on small pelagic fish.

Following the introduction, Section 2 elaborates the materials and methods including the conceptual framework, method of data analysis, as well as some demographic characteristics of the research population. Section 3 is devoted to the results and discussions of the survey. Section 4 concludes the paper and the final section makes some recommendations for policy consideration.

## 2. Materials and Methods

### 2.1. Conceptual Framework

According to the Food and Agriculture Organization (FAO) [11], "food security exists when all people, at all times, have physical and economic access to sufficient, safe and nutritious food that meets their dietary needs and food preferences for an active and healthy life". In line with this perspective, the scholarly and practitioner communities have identified four dimensions of food security, highlighting (1) the availability of sufficient quantities of food; (2) the capacities of people to access (purchase) this food; (3) the quality of food and its utilization; and (4) the continued access to food over time (stability) [12].

Over the years, food and nutrition security has become an integral part of a more complex food systems approach, encompassing the production, processing, packaging, transporting, marketing, consumption, and disposal of food [13]. In line with this trend, the High Level Panel of Experts on Food Security and Nutrition [14] of the Food and Agriculture Organization (FAO) linked food systems, and food security to the Sustainable Development Goals. In this context, the four dimensions of food security were then reduced to three: (1) food availability (proximity), (2) economic access (affordability), and (3) food quality and safety. However, since Ingram [15] emphasized that "stability", or continuity over time appears to be seen as a factor essential to each of the three dimensions of food security

outlined by the High Level Panel of Experts [14], this paper adheres to the four dimensions as drafted by the Food and Agriculture Organization (FAO) [11].

The four dimensions of food and nutrition security are outputs (end-of-pipe) of the workings of the food system that can be calculated at individual, household, or other aggregate levels. The nature of the food supply chain is one important determinant of food and nutrition security, as are a number of other concerned drivers (related to the environment, technology and infrastructure, politics and economics, socio-cultural features, and demographic trends), see [14]. Obiero et al. [16] note that food and nutrition security is influenced by consumer behaviour: the choices consumers make regarding what food to purchase, prepare, cook, store, and eat. These choices follow from the specific characteristics of households, and crystallize into a certain number of consumption patterns. Figure 1 presents the conceptual framework as adapted for the purpose of this paper.

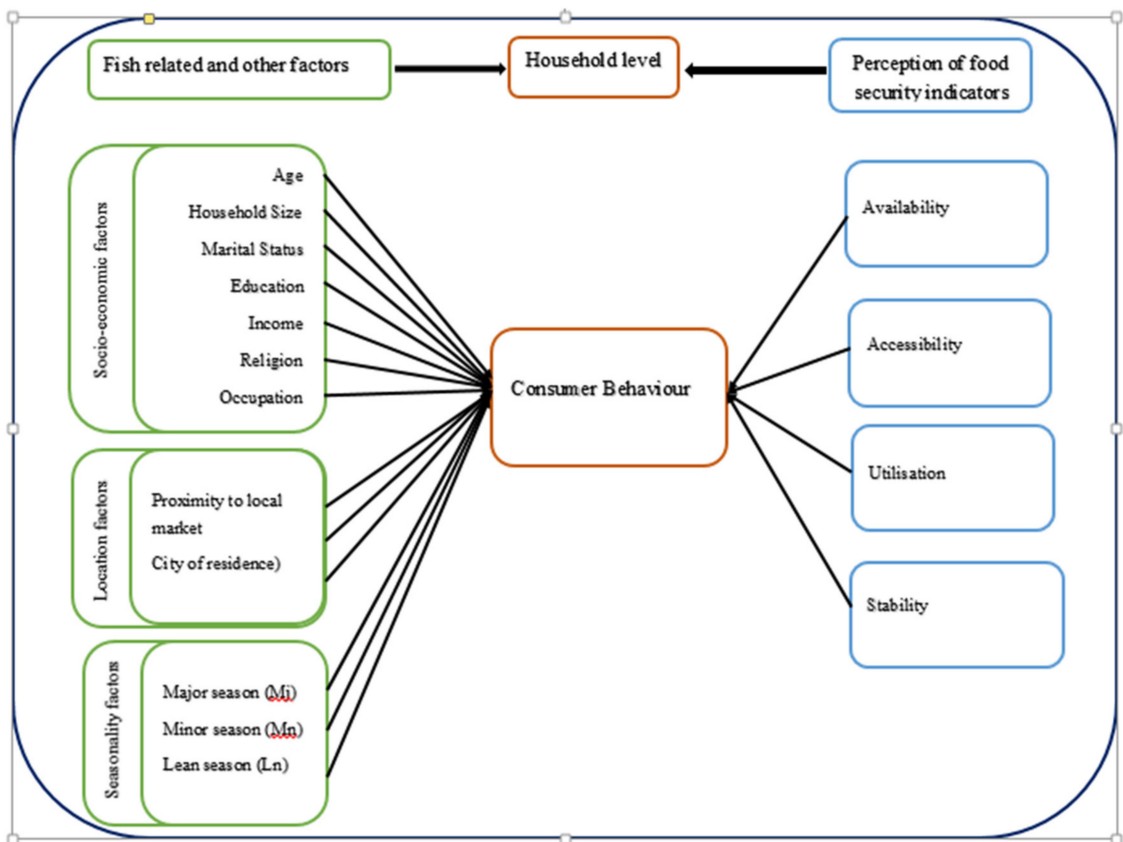

**Figure 1.** Conceptual framework of consumer behavior and perceptions of fish food security in poor urban households in Ghana.

The paper assumes that given the amount of money allocated to fish consumption out of the total food expenditure, the consumer is influenced by socio-economic, location, and seasonality factors. The socio-economic factors of interest include age, household size, marital status, educational level, income, religion, and occupation. Location factors include proximity to local market and city of residence. Seasonality factors include the season of marine fish catch (major, minor, and lean seasons). Drawing from Obiero et al. [16], the paper further notes that the perception of food security may influence the fish consumption behaviour of households. The perception of fish food security is operationalized by using the four food security dimensions: availability ("I get fish to buy when I need it"), accessibility ("I believe the fish prices are generally affordable"), utilization/quality ("Good quality fish is available for me to purchase"), and stability ("I get the type(s) of fish I want all throughout the year").

### 2.2. Study Area and Sampling Technique

A multistage sampling approach was adopted for the study where Accra (coastal city) and Tamale (inland city) were purposely selected in the first stage to capture geographic location differences. In the second stage, four poor neighbourhoods were selected in both cities based on literature on the distribution of endemic poverty areas [17,18] and assuring a geographical spread. the neighborhoods of Accra selected are Nima, Chorkor, Ga Mashie, and James Town; and those of Tamale are Sagnarigu, Kukoo, Sakasaka, and Salamba. These suburbs are all characterized by spatially unplanned settlements with high population densities. Infrastructure in these suburbs is often inadequate and dwellings are constructed with poorer materials [19]. In the third stage, a simple random sampling technique was employed to select 31–39 households within each of the selected neighbourhoods. About two minutes' walking distance between households was ensured for a good spread of interviewed households within any one neighbourhood. The survey gathered information on each household's socio-economic characteristics, fish types consumed, price and source of fish, income allocation to fish consumption, and perceptions of fish food security.

As a first step in the data collection, a pilot survey was carried out to validate the suitability, appropriateness of the questions and expected responses. The questionnaire was then revised in light of errors detected from the pilot survey. Data collection was repeated three times in a one-year timespan during the 2017/2018 fishing season to take into consideration seasonality effects on fish catch. Efforts were made to obtain reliable information from the same households during the three times survey with average success rate of 93.2% in identifying same households. Two hundred and ninety three (293) households took part in the first survey effort. This number was reduced in the subsequent surveys to 279 and 267 households, respectively, in the second and the third surveys, as a result of difficulty in finding the same respondents. Finally, a total of 839 responses were considered for the analysis. The first round of data collection was done in November, whilst the second and third surveys were conducted in March and August, respectively, in the following year to capture seasonality effects (minor, lean, and major peak landing seasons) on income allocation to fish consumption. A summary of survey distribution of sample sizes of respondents in the various neighborhoods is presented in Table 1.

**Table 1.** Distribution of sample sizes of respondents in the selected neighbourhoods of Accra and Tamale.

| City | Suburb | 1st Round (Minor Season) | 2nd Round (Lean Season) | 3rd Round (Major Season) |
|---|---|---|---|---|
| Accra | Nima | 35 | 34 | 32 |
| | Chorkor | 38 | 35 | 34 |
| | Ga-Mashie | 39 | 37 | 35 |
| | James Town | 34 | 33 | 32 |
| Tamale | Sagnarigu | 39 | 35 | 34 |
| | Kukoo | 36 | 35 | 34 |
| | Sakasaka | 38 | 37 | 35 |
| | Salamba | 34 | 33 | 31 |
| Total | | 293 | 279 | 267 |

### 2.3. Data Analysis

This paper adopted the modified Cobb–Douglas regression model to analyze the determinants of household expenditure on fish consumption, whilst considering a 1–5 Likert scale and descriptive statistics to analyze the perception of fish food security in the study area.

#### 2.3.1. Modified Cobb–Douglas Model Specification

In analysing the determinants of economic models, studies often adopt the Cobb–Douglas regression techniques to assess the effect of various exogenous factors on the dependent variable.

However, the conventional Cobb–Douglas approach fails to consider the influence of non-continuous variables on economic models in a log transformed perspective. As a result, this paper considered a modified Cobb–Douglas function which has also been used in other studies [20,21] to analyze factors that influence the allocation of income to fish purchase whilst accounting for categorical exogenous factors. Since the dependent variable in this paper (log of money spent on fish per week) is a continuous variable with a mixture of continuous and categorical determining factors, an econometric model using a modified Cobb–Douglas expenditure function may produce robust, optimal, and consistent estimates [21]. The general Cobb–Douglas production assumes the form:

$$Y = A X_1^{\beta_1} X_2^{\beta_2} \dots X_n^{\beta_n} \tag{1}$$

As proposed by the earlier studies [20,21], a modified Cobb–Douglas expenditure function is expressed as:

$$Y = A X_1^{\beta_1} X_2^{\beta_2} \dots X_n^{\beta_n} e^{k_1 Z_1 + k_2 Z_2 + \dots + k_n Z_m} \tag{2}$$

where

$$Z = \left\{ \begin{array}{c} 1 \\ 0 \end{array} \right. , \; Dummy \; Variabe \tag{3}$$

In the context of this paper, the modified Cobb–Douglas is expressed as:

$$
\begin{aligned}
ln(Y_i) = \; & \beta_0 + \beta_1 ln(Age_{1i}) + \beta_2 ln(Household\ size_{2i}) + \beta_3(Marital\ status_{3i}) \\
& + \beta_4(Income_{4i}) + \beta_5(Education_{5i}) + \beta_6(Religion_{6i}) \\
& + \beta_7(Occupation_{7i}) + \beta_8(Proximity\ to\ local\ market\ _{8i}) \\
& + \beta_9(City\ of\ residence_{9i}) + \beta_{10}(Major\ season_{10i}) \\
& + \beta_{11}(Minor\ season_{11i} \\
& + \beta_{12}(Interaction\ of\ religion\ and\ major\ season_{12i}) \\
& + \beta_{13}(Interaction\ of\ religion\ and\ minor\ season_{13i}) + \varepsilon_i
\end{aligned} \tag{4}
$$

where $i = 1, 2, \dots , 839$, representing the number of observations, $Y_i$ is the dependent variable representing the log of actual money in Ghana Cedis spent on fish per week by households (fish consumption); $x_i$ is the vector of factors explaining the variation of household money allocation to fish consumption; $\beta$ is the vector of parameters to be estimated; and $\varepsilon_i$ is error term assumed to be independently normally distributed as $\varepsilon_i \sim N(0,\sigma^2)$.

2.3.2. Operationalization of Model

The dependent variable in the modified Cobb–Douglas regression model is defined as the amount of money (GHS) spent on fish per week, and is given as household expenditure on fish. This paper used the following explanatory variables in the modified Cobb–Douglas function to explicate the variation in amount of money spent on fish per week: age, household size, marital status, education level, income, religion, occupation, proximity to local market, city of residence, and trend variables to capture seasonal effect. The study further considered the interactive effect of religion and seasonal fish catch on the expenditure of fish in the study area. Descriptive summary of the dependent variable and the explanatory variables are presented in Table 2.

**Table 2.** Descriptive summary of determinants of food income allocation to fish consumption.

| Variable | Categorization | Frequency | Percent | Cumulative |
|---|---|---|---|---|
| Marital status | Married | 645 | 76.88 | 76.88 |
| | Otherwise | 194 | 23.12 | 100 |
| Education | No formal education | 328 | 39.09 | 39.09 |
| | primary education | 68 | 8.1 | 47.2 |
| | secondary education | 216 | 25.74 | 72.94 |
| | polytechnic education | 188 | 22.41 | 95.35 |
| | tertiary education | 39 | 4.65 | 100 |
| Household head income per month | less than 100 | 37 | 4.41 | 4.41 |
| | 101–200 | 366 | 43.62 | 48.03 |
| | 201–500 | 234 | 27.89 | 75.92 |
| | 501–1000 | 109 | 12.99 | 88.92 |
| | 1001–2000 | 79 | 9.42 | 98.33 |
| | more than 2000 | 14 | 1.67 | 100 |
| Religion | Christian | 363 | 43.27 | 43.27 |
| | Muslim | 476 | 56.73 | 100.00 |
| Occupation | Formal | 365 | 43.50 | 43.50 |
| | Otherwise | 474 | 56.50 | 100 |
| Proximity to local market | Local market | 559 | 66.63 | 66.63 |
| | Otherwise | 280 | 33.37 | 100 |
| City of residence | Accra | 418 | 49.82 | 49.82 |
| | Tamale | 421 | 50.18 | 100 |
| Seasonality of fish | Major season (Mj) | 293 | 34.92 | 34.92 |
| | Minor season (Mn) | 278 | 33.13 | 68.05 |
| | Lean season (Ln) | 268 | 31.95 | 100 |
| Interaction of religion and major season | Religion*major season | 148 | 17.64 | |
| Interaction of religion and minor season | Religion*minor season | 120 | 14.30 | |
| | **Mean** | **Std. Dev.** | **Min** | **Max** |
| Money spent on fish per week | 31.149 | 22.487 | 5 | 203 |
| Age | 36.882 | 10.611 | 19 | 80 |
| Household size | 11.39 | 9.39 | 1 | 55 |

### 2.3.3. Perception Analysis

Means and frequencies were used to analyze the consumers' perception of fish consumption and fish food security in the study area. Consumers' evaluation of perception was done by using a simple 1–5 Likert scale (1 = strongly agree; 2 = agree; 3 = neutral; 4 = disagree; 5 = strongly disagree). The Weighted Mean Index (WMI) was adopted to analyze the responses from the five-point Likert scale given in Equation (5). The Likert scale was considered to measure the extent to which consumers agree or disagree with the perception of food safety for fish.

$$\text{WMI} = (Q_i * W_i)/N \tag{5}$$

where: $Q_i$ is the response rate to the *ith* factor, $W_i$ is the weight of the *ith* factor, and $N$ is the overall number of responses.

## 3. Results and Discussions

The findings of the study in terms of the types of fish consumed by households, frequency of fish consumption, the determinants of household expenditure allocation to fish consumption, and perception analysis of fish food security are discussed in this section.

### 3.1. Types of Fish Consumed

The species of fish consumed by low-income households in the poor urban neighborhoods are presented in Table 3.

**Table 3.** Summary of various varieties of fish consumed in the suburbs of Accra and Tamale.

| Fish Species | Round 1 (%) | Round 2 (%) | Round 3 (%) | Pooled (%) |
|---|---|---|---|---|
| | November | April | July | All Year Round |
| Mackerel *(Salmon)* | 38.6 | 32.3 | 20.9 | 31.3 |
| Sardinella | 28.0 | 33.8 | 22.9 | 28.7 |
| Sea Bream | 9.3 | 9.2 | 17.1 | 11.5 |
| Atlantic horse mackerel | 10.0 | 8.7 | 8.1 | 9.0 |
| Barracuda | 9.7 | 6.7 | 9.4 | 8.5 |
| Redfish | 7.4 | 8.4 | 6.4 | 7.5 |
| Anchovy | 6.9 | 7.2 | 6.2 | 6.8 |
| Tuna | 5.5 | 5.9 | 5.8 | 5.7 |
| Tilapia | 6.2 | 3.4 | 7.7 | 5.6 |
| Burrito | 4.5 | 5.7 | 3.0 | 4.5 |
| Catfish | 3.6 | 2.8 | 4.9 | 3.7 |
| Others | 8.7 | 8.2 | 8.5 | 8.5 |
| Total | 100 | 100 | 100 | 100 |

Among the thirty-four (34) fish species obtained from the survey, the most consumed fish is mackerel (locally known as *salmon*) which is reported by 31.3% of households interviewed. This is followed by the sardinella (*amane/eban*) with 28.7%, seabream with 11.5%, Atlantic horse mackerel (9.0%), and anchovy (6.8%). Consistent with findings of other studies [22–25], this paper demonstrates that small pelagic fish species such as mackerel, sardinella, and anchovies are more frequently consumed by poor households compared to large pelagic species such as tuna and demersal species such as cassava fish, which are considered relatively expensive.

Even though the study revealed that seabream is the third most preferred fish species in the study area, the United States Agency for International Development (USAID) [26] reported that seabream, red snapper, and cassava fish are preferred in Ghana but they are relatively expensive. Unlike inland areas like Tamale, these species are rather more popular among coastal populations who are fish-eating and will spend more to meet this utility. Onumah et al. [27] noted that fish farming in Ghana (mainly tilapia and catfish farming) is being promoted by the government to mitigate the gap between fish production and consumption in the country. However, contrary to this national endeavour, tilapia and catfish were identified to be 9th and 18th most preferred fish, respectively, probably due to their expensive nature.

### 3.2. Frequency of Fish Consumption

The frequency of fish consumption per week—whether marine fresh, inland fresh (aquaculture or freshwater) fish, or smoked (marine and inland) fish, is presented in Table 4.

**Table 4.** Frequency of fish consumption among households in Accra and Tamale.

| Row Labels | Marine Fresh Fish (%) | | | Inland Fresh Fish (%) | | | Smoked Fish (Inland and Marine) (%) | | |
|---|---|---|---|---|---|---|---|---|---|
| | Accra | Tamale | Pooled | Accra | Tamale | Pooled | Accra | Tamale | Pooled |
| 6–7 days a week | 74 | 11 | 42 | 47 | 6 | 26 | 19 | 11 | 15 |
| 4–5 times a week | 4 | 25 | 15 | 2 | 6 | 4 | 4 | 41 | 22 |
| 1–3 times a week | 8 | 35 | 22 | 9 | 26 | 18 | 26 | 38 | 32 |
| 1–3 times per month | 7 | 11 | 9 | 16 | 29 | 22 | 11 | 10 | 10 |
| Less than monthly | 6 | 11 | 8 | 26 | 22 | 24 | 31 | 1 | 16 |
| Never | 1 | 9 | 5 | 0 | 12 | 6 | 9 | 1 | 5 |
| Grand Total (%) | 100 | 100 | 100 | 100 | 100 | 100 | 100 | 100 | 100 |

A greater percent of households in Accra purchased marine fresh fish (74%) as compared to 11% in Tamale for consumption between 6 and 7 times within a week. In the case of Tamale, a greater percent of households (41%) responded that they purchase smoked fish for consumption between 4 and 5 times within a week as compared to 4% of households in Accra. Consistent with the findings of Gordon et al. [28], fresh marine fish is predominantly consumed in the coastal cities of Ghana, including Accra, probably due to closer proximity to the sea than smoked fish which is highly patronised in inland cities such as Tamale. The study further revealed that 47% of households in Accra purchased inland water fresh fish (largely fresh farmed fish) as compared to 6% of households in Tamale. Akuffo and Quagrainie [29] noted that freshwater farmed fish are comparatively more expensive than marine fresh fish and since the poverty status of households in Accra is relatively lower than in Tamale [30], households in Accra are likely to consume more inland fresh fish than households in Tamale. Moreover, Accra (a city in the south) is a major target market for the majority of the inland farmed fish predominantly produced in southern Ghana, mainly on the Lake Volta and other areas. Even though a number of organisations and scholars [1,5,10] have stated that fish consumption is a major component of the diet of most Ghanaians; on average, 5%, 6%, and 5% households from both cities reported that they had never purchased marine fresh fish, inland fresh fish, and smoked fish, respectively, for consumption. These are likely to be vegetarian or meat consuming households who do not include fish in their diets. Chagomoka et al. [31] mentioned in their research on vegetable production and consumption and its contribution to diets along the urban–rural continuum in northern Ghana that vegetarian households are gradually increasing in cities of Ghana as a results of health concerns.

*3.3. Determinants of Household Expenditure on Fish Consumption*

The average household weekly expenditure on fish consumption is estimated in this current study to be GHS 31.15 (Euro 4.94 at 0.16), as presented in Table 2. This result is slightly smaller than the national mean weekly household expenditure of GHS 35.54 (Euro 5.64) on fish and sea food

consumption, which represents a 15.8% share of the food budget [32]. The national mean weekly household expenditure on meat was noted to be GHS 15.05 (Euro 2.39), representing a food budget share of 7.6%, whilst that on egg and dairy products (milk and cheese) was calculated to be GHS 4.53 (Euro 0.72), representing 3.0%. The Ghana Statistical Service [33] also documented the share of fish, meat, and egg and dairy products (milk and cheese) in the food budget to be 16.4%, 7.4%, and 2.6%, respectively. The results of this study and that of the Ghana Statistical Service (GSS) underscore the fact that fish form a significant part of the diets of households in Ghana.

Results of the modified Cobb–Douglas function demonstrating the effects of variables influencing household expenditure on fish consumption are presented in Table 5. The findings revealed that all the variables except age, household size, and education significantly influenced household fish expenditure.

**Table 5.** Modified Cobb–Douglas estimates of effects of some variables influencing household expenditure on fish consumption.

| Variables | Coefficient | Standard Rrror | P > |t| |
|---|---|---|---|
| Ln Age | −0.001 | 0.065 | 0.985 |
| Ln Household Size | 0.011 | 0.026 | 0.688 |
| Marital status | 0.070 * | 0.041 | 0.091 |
| Income | −0.086 *** | 0.015 | 0.000 |
| Education | 0.005 | 0.015 | 0.724 |
| Religion | 0.400 *** | 0.073 | 0.000 |
| Occupation | 0.146 *** | 0.035 | 0.000 |
| Proximity to local market | 0.094 ** | 0.039 | 0.015 |
| City of residence | 0.357 *** | 0.059 | 0.000 |
| Major season (Mj) | −0.272 *** | 0.056 | 0.000 |
| Minor season (Mn) | −0.470 *** | 0.053 | 0.000 |
| Religion * Mj | −0.200 ** | 0.086 | 0.020 |
| Religion * Mn | −0.330 *** | 0.085 | 0.000 |
| Constant | 3.269 | 0.275 | 0.000 |

| Statistical Parameter | Value |
|---|---|
| Observations | 839 |
| Prob > F | 0.000 |

*** $p < 0.01$, ** $p < 0.05$, * $p < 0.1$ significant levels; Ln = Natural log; Mj = Major season; Mn = Minor season.

This paper found a negative but insignificant relationship between age of consumers and amount of money spent on fish per week as seen in Table 5. Household size has a positive but insignificant effect on expenditure on fish consumption within the week. This may imply that a household of a larger size may need more fish to meet its fish consumption requirements and hence may spend more to acquire large quantities of fish.

Marital status is found to have a significant positive influence on fish expenditure at 10%. This means that married respondents are found to spend more on fish consumption. The result from the modified Cobb–Douglas regression analysis indicates that being a couple increases the expenditure on fish by 0.070%. Using the additionality effect, being married implies additional mouths to eat fish, thereby increasing fish consumption expenditure. This corroborates the finding in Ethiopia where a household head being married increases the consumption of some food products [34].

As expected, household income is negative and statistically significant at 1% level. This indicates that an increase in household income by one Cedis would result in 0.086% reduction in the amount of money spent on fish. The outcome confirms the assertion that an increase in income would lead to a decline in household expenditure on fish. This result supports the argument that the rich spend a smaller share of their food income on fish than the poor who may allocate a bigger share to fish [10]. Intuitively, an increase in income allows part of the household fish budget to be spent on other goods like beef, goat meat, mutton, and other luxury protein foods, as also reported in a study on

Lusaka residents in Zambia [35]. Liu et al. [36] note that increase in income led to increase in beef expenditure compared to fish consumption in some provinces in China and increase in teff consumption in Ethiopia [34].

The education variable is estimated to be positive but statistically insignificant. This may imply that the increasing education level of respondents may increase the amount of money households spend on fish consumption. This contrasts the findings of other studies [37,38], claiming that people with higher education are more likely to increase consumption of animal products. The positive results obtained in our survey might be due to reduction in dependence on bush meat and animal protein in general because of the outbreak of Ebola and recent health campaigns on the health benefits of fish in Ghana.

This study further found that religion has a positive and significant effect on household expenditure on fish purchase. Christians in our sample spent about 0.4% more on fish consumption than Muslims. Muslims have some restrictions on fish consumption. Gadegbeku et al. [39] noted that Muslims normally refrain from eating non-scaled fish, unlike Christians. This finding corroborates the assertion that many early Christians in Europe, North America, China, India, and parts of Africa prefer fish in their diets [40].

The occupation of the consumer is revealed to have a positive and significant effect on the amount spent on fish purchase in the study area. Those respondents who have formal jobs spent about 0.146% more on fish purchase than those in informal employment. Formal workers may be earning a regular income, and this could guarantee a constant amount of money allocated for household food purchase including fish. The Food and Agriculture Organization (FAO) [41] suggested that due to higher food prices during the world food crisis of 2006–2008, formal workers who received regular source of income may spend more on food, including fish, than the informal workers.

Furthermore, proximity to local market has a positive and a significant relationship with household expenditure on fish consumption. An increase in the number and density of markets increased accessibility of the fish by 0.094%. A consumer in making a choice for protein can therefore make a quick dash to a nearby market irrespective of other factors to buy fish. This may translate into an increased amount of money spent by households on fish. The feeling that such fish is better in terms of freshness may account for this outcome. Among other factors such as region, employment, household size, and income, it was observed that proximity to local markets significantly affected the likelihood of eating fish and shellfish in the United States [42].

The city of residence variable is estimated to be positive and statistically significant at the 1% level. The results demonstrate that households residing in Accra are likely to allocate a higher percentage of their income to fish consumption than their counterparts in Tamale. The estimated coefficient suggests that households in Accra are likely to spend 0.357% more on fish consumption than households in Tamale. Marine fish is cheaper than animal products such as beef in Accra. However, most consumers in Tamale consume the cheaper species such as anchovies and therefore may spend a smaller percentage of their income to buy fish. Additionally, in northern Ghana, people are also more oriented toward the rearing of livestock [43] and therefore use fish to supplement meat in many cuisines, thereby reducing their expenditure.

The seasonality effect of fish catch on the amount of money spent on fish consumption is found to be negative and statistically significant at 1%. These findings generally agree with the fact that households spend less on fish consumption in the major season (Mj) and minor season (Mn), respectively, as compared to the amount spent on fish during the lean the season. This may be argued on the grounds that even though Ghana relies on about 50% fish imports to augment domestic production [5], increase in fish catch during the bumper and minor seasons, as compared to the lean season, contribute a lot in the reduction of fish prices which eventually causes a reduction in the amount of money allocated to fish consumption in the study area. Fish prices are generally reduced in Ghana during the bumper season [22]. However, the findings on the effects of the seasonal fish catch on household fish expenditure show that it is rather smaller in the minor season (−0.470) than in

the major season (−0.272). To understand these results, the study further analyzed the interaction of religion and season and obtained a significant coefficient of −0.200, whilst the interaction of religion and minor season is estimated to be −0.330 at 1% significant level. These findings suggest that religious celebrations corroborate with season of fish catch to reduce the amount of money spent on fish consumption. The minor season coincides with Christmas festivities. During this period, Christians spend relatively more money to purchase chicken, beef, and other products like eggs and mushroom than fish products [44]. The major season, however, coincides with the celebration of Eid al-Adha in July where Muslims celebrate the occasion predominantly with the consumption of meat [45].

### 3.4. Perception of Fish Food Security

Household perceptions of food security and nutritional status are discussed from the data in Table 6 in terms of the four dimensions of food security mentioned in Section 2.1.

### 3.4.1. Availability

Individuals require sufficient quantities of appropriate food to be available from domestic production or commercial imports. As revealed by the study, about 87% of the households either strongly agree (47%) or agree (40%) that fish is available when needed. Only 4% strongly disagree that fish is obtainable when needed. This result corroborates the findings that there is a wide variety of fish available in Ghana's markets [10]. In Ghana, a number of traditional practices including smoking are adopted, especially in the major season, to increase the shelf life of fish. These processed fish can then be transported far from the coastal regions. Markets are established across the fish value chain involving processors and traders who supply and distribute processed fish along dispersive chain networks, ensuring availability in non-producing centres. Additionally, traders who have large storage facilities buy and store both local and imported fish and release them unto the market in the lean season, encouraging availability of fish for households [28].

### 3.4.2. Accessibility

A sufficient amount of food (fish) is required by individuals in order to be food secure. The relative increase in the supply of fish products is a necessary but not a sufficient condition for food security, since other factors such as economic assurance and the social and physical access to fish are also important [46]. According to Altenburg [47], there is a high degree of competition in the small pelagic chains which allows trading activities or powerful traders to determine prices. It can be observed from Table 6 that households have varied opinion on accessibility (affordability) of fish. Whilst about 43% of households either strongly agree or agree that fish prices are generally affordable, about 45% of the households either disagree or strongly disagree on the issue of fish affordability. Also, households in Accra (32%) and Tamale (40%) disagree about the notion that fish is generally affordable. However, 12% of households are neutral in their responses. In general, households are of the view that the high volatility of fish prices on the market due to seasonality could affect fish access. Hence, where households could not afford to buy fish due to high prices, they tend to substitute it with other protein products.

**Table 6.** Consumers' perception of fish consumption and fish food security.

| Perception | Availability: I Get Fish Supplied to Me When I Need It (%) | | | Accessibility: I Believe the Fish Prices are Generally Affordable (%) | | | Quality: Good Quality Fish Is Available for Me to Purchase (%) | | | Stability: I Get the Fish I Want Throughout the Year (%) | | |
|---|---|---|---|---|---|---|---|---|---|---|---|---|
| | **Accra** | **Tamale** | **Total** | **Accra** | **Tamale** | **Total** | **Accra** | **Tamale** | **Total** | **Accra** | **Tamale** | **Total** |
| Strongly agree | 46 | 47 | 47 | 2 | 8 | 5 | 33 | 33 | 33 | 32 | 24 | 28 |
| Agree | 44 | 36 | 40 | 44 | 31 | 38 | 48 | 46 | 47 | 45 | 38 | 42 |
| Neutral | 1 | 4 | 2 | 14 | 10 | 12 | 4 | 6 | 5 | 5 | 10 | 7 |
| Disagree | 5 | 10 | 8 | 32 | 40 | 36 | 10 | 13 | 12 | 13 | 23 | 18 |
| Strongly Disagree | 4 | 4 | 4 | 8 | 11 | 9 | 5 | 2 | 3 | 6 | 6 | 5 |
| Grand Total | 100 | 100 | 100 | 100 | 100 | 100 | 100 | 100 | 100 | 100 | 100 | 100 |

Source: Survey Data.

### 3.4.3. Utilization

An important aspect of food security is the effective and efficient utilization of food resources. It is believed that dealers in the fish value chains have a strong interest in supplying quality fish products to customers in order to maintain/improve their reputation and dependability. Notwithstanding, Sakyi et al. [48] observed that fish quality could be compromised due to poor (unhygienic) handling practices, poor processing technologies, and a bad marketing mechanism. Others [49,50] speculated that chemical adulteration during fishing, storing, and preserving could be practiced along the fish value chain. Additionally, the use of non-standardized packaging materials during fish transportation and marketing [22] and open exposure of fish products to the weather at the market place could lead to loss of economic and nutritional value of fish, thereby compromising fish quality [51]. Contrary to these findings, about 80% of households either strongly agree or agree that good-quality fish is available for purchase in the study area. This finding suggests that in spite of the speculated issues of poor-quality fish, on average, the majority of households in the study area are satisfied with the quality of fish they consume.

### 3.4.4. Stability

In order for a population, household, or individual to be food secure, they must have access to adequate food (fish) at all times (stability). Thus, fish must be available all year round, regardless of the economic situation being faced. The finding of this study reveals that the majority of households from both cities (70%) either strongly agree or agree to being able to get the fish they wanted all year round. Coupled with available cold storage facilities which ensure marketing and distribution of frozen fish, Nunoo et al. [22] supported the assertion that processors with large smoking and storage capacities are able to smoke to ensure supply during the lean and fish deficit seasons. These mechanisms ensure availability of fish throughout the year and also help stabilize fish prices to some extent.

## 4. Conclusions

This paper assessed the fish consumption behaviour and the perception of fish food security in terms of availability, accessibility, utilization, and stability to low-income households residing in poor urban neighbourhoods in coastal and inland Ghana. The paper focused on two cities in Ghana and ignores the rural–urban dynamics to highlight differences in fish consumption between rural and urban areas. It also examined the fish consumption behaviour of households in different income strata, especially those in the lowest income quintile. The paper considered three season survey data to capture the effect of seasonality. The modified Cobb–Douglas regression model was used to analyze the determinants of household food expenditure on fish consumption, whilst a 1–5 Likert scale and descriptive statistics were adopted to analyze the perception of fish food security in the study area.

The results demonstrate that small pelagic fish species such as mackerel, sardinella, and anchovies are commonly consumed compared to large pelagic and farmed species, which are considered relatively expensive by the households. Findings on the consumption trend in marine fresh, freshwater and smoked fish reveal that a greater percent of households in Accra purchased marine fresh fish, whereas a greater percent of households in Tamale consumed smoked fish. These results demonstrate that proximity to the sea influences the state in which fish is consumed.

The mean expenditure on fish consumption per week estimated underscores that fish forms a significant part of the diets of households in Ghana. Results further show that marital status, education, religion, occupation, proximity to local market, and household living in Accra had a positive and significant influence on household expenditure on fish consumption, whilst level of income had a significant negative effect. Households spend less on fish consumption in the bumper season and minor season compared to amount of money spent on fish during the lean season. Moreover, Christians and Muslims often spend less money on fish consumption, especially during festive occasion such as Christmas and Eid al-Adha celebrations.

Finally, the paper revealed that whilst fish is readily available and can be obtained throughout the year (stability), households are unable to purchase the fish they would like to consume possibly because of increased prices. Nonetheless, households from both cities had varied opinion on accessibility (affordability) of fish probably due to high volatility of fish prices on the market as a result of seasonality. Notwithstanding the opinion in the literature that fish quality could be compromised due to poor (unhygienic) handling practices, the majority of households either strongly agree or agree that good quality fish is available for purchase in the study area.

## 5. Suggestions for the Future

Based on the findings from the study, the paper recommends that the Ghanaian government should support and strengthen value chains for small pelagic fish species such as sardinella, mackerel, and anchovies since they are preferred by poor households. All stakeholders (including government, processors, and traders) should strive to put in place mechanisms to improve transport and distribution systems that will provide consumers with the best quality fish, and encourage policy reforms to help reduce fish price volatility.

**Author Contributions:** E.E.O. helped in the design of the study, data collection, data analysis using STATA, and write-up of the manuscript. E.A.Q. assisted in the data analysis and write-up of the paper. A.K.A. helped in the data collection, data analysis, and write-up of the manuscript. B.B.C. assisted in the study design, data collection, and write-up of the paper. All authors have read and agreed to the published version of the manuscript.

**Funding:** This publication is made possible by the generous support of the Netherlands Organization for Scientific Research (NWO project W 07.50.1818). The contents are the responsibility of the authors as part of the Fish for food security in city regions: an inter-regional innovation project and do not necessarily reflect the views of the NWO.

**Acknowledgments:** We are grateful to Maarten Bavinck and Scholtens Joeri from the University of Amsterdam, the Netherlands, Amalendu Jyotishi for their insightful suggestions and reading through the manuscript.

**Conflicts of Interest:** There are no conflicts of interest to declare.

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
