# Peer review of "Fish Consumption Behaviour and Perception of Food Security of Low-Income Households in Urban Areas of Ghana"

_sustainability, doi:10.3390/su12197932_

Round 1
Author Response
Please find point-by-point responses to comment in the file attached

Reviewer 2 Report
The paper is interesting, but it's very specific for the country fo Ghana.
I think it's important to specify how we can use the results also in other countries and what the paper add to the previous papers in the internationa literaure.
the limits of the reseach are missed. I recommend to add them in the conclusion section.
Author Response
Please find point-by-point response to commemts in the attached file.

Reviewer 3 Report
The work is of interest for two important reasons: 1. The impact that local fishing has on the country's economic structure, which is evaluated at 5% of GDP, according to data provided by the author. 2. Fish as an essential ingredient for consumption in the different regions of Ghana, with consumption rates higher than the African average.
Given this, I think some considerations are important: a) If it would be possible to calculate the nutrient and caloric contribution of this important consumption of small species fish for the young population. Know how many kilocalories we are talking about daily consumption, and what relevance the fish has in them. b) Incorporate (I do not think it was very complicated) some indicator of life expectancy of the population, in contrast to other African countries. Perhaps, if this were possible, perform some regression analysis between life expectancy and fish consumption, alongside GDP growth. c) It would be important to know if these activities can have any way out beyond self-consumption; that is, if they could generate exports and, therefore, income that could improve the quality of life of the population. These recommendations are provided that the sources allow to address the issues I have proposed. I consider that the work is publishable and, above all, I have found it of great interest that concrete recommendations be made to the government. Science must provide solutions to politics, especially in current times.
Author Response
Please find poin-by-point response to comments attached.

Reviewer 4 Report
The authors have elaborated an interesting work.
In my opinion, the documentation, regarding Literarature Review, should be extended with titles from WoS magazines (10-11) papers.
However, strictly apply the editorial rules of the magazine.
The writing of the paper presents multiple negligences (Ex):
”Fisheries is one of the key sectors supporting the socio-economic development of Ghana. [1] note
30 that the sector generates about US$ 1 billion in revenue each year and supports about 135,000 fishers in the marine sub-sector alone.” (L29-31).
”Aquaculture is also progressively gaining grounds in contributing to the domestic market. However, farmed fish is relatively expensive and Eventually, be reviewed by a native (English) teacher. may be unaffordable to poor consumers [6]. [1] reveal that the poor are known to depend largely on low-price, small pelagic fish species, such as sardines and mackerel which are processed in a variety of ways....”
Obviously, the authors will also be able to find other errors ... that they need to correct.
Attention! including how to play References.
Eventually, be reviewed by a native (English) teacher.
Author Response
Please find point-by-point response to comments attached.

Round 2
Reviewer 4 Report
I found that the work was wonderful.
Consequently, I appreciate that it is publishable.